# Staphyloxanthin-enriched extracts promote biofilm formation and oxidative stress resistance in *Staphylococcus aureus*

Jingyi Yu,[1] Li Shen,[1] Jinjin Yang,[1] Junhong Shi,[1] Yu Huang,[2] Yongpeng Shang,[1] Fangyou Yu[1]

**ABSTRACT** *Staphylococcus aureus*, an opportunistic pathogen of global health concern, presents a significant clinical challenge due to its escalating antibiotic resistance and biofilm-forming capacity. The biofilm matrix of *S. aureus* is enriched with carotenoids, primarily staphyloxanthin (STX), which function as virulence factors by scavenging reactive oxygen species and inhibiting antimicrobial peptides. In this study, we examined the impact of the methanol extract of *S. aureus* (MES) on biofilm formation. Our findings revealed that MES enhanced biofilm formation in *S. aureus* strains with inherently weak biofilm-forming ability by upregulating key adhesion genes (fibronectin-binding protein A/*fnbB*, serine-aspartate repeat-containing protein D, clumping factors A/B, elastin-binding protein, and *fib*) and downregulating autolysis-associated genes (*lytR* and *lrgA*). Furthermore, MES augmented the resistance of these strains to whole blood-mediated killing and improved their antioxidant capacity. To elucidate the role of STX, methanolic extracts were prepared from *crtM* and *crtN* mutants of the USA300 LAC strain and applied to biofilm-impaired strains. These experiments provided indirect evidence that STX in the methanolic extract is a critical mediator of biofilm promotion *in vitro*. Collectively, our results suggest a potential mechanistic link between STX in *S. aureus* methanolic extract and biofilm formation, offering novel insights for therapeutic strategies against *S. aureus* infections.

**IMPORTANCE** Our findings demonstrate that the methanolic extract of *S. aureus*, predominantly comprising STX, augments biofilm formation and antioxidant capacity *in vitro*. These results not only offer novel insights into potential therapeutic strategies for *S. aureus* infections but also underscore the potential role of microbial secondary metabolites in interstrain interactions.

**KEYWORDS** methicillin-resistant *Staphylococcus aureus*, biofilm, staphyloxanthin, carotenoids

$\mathcal{S}$taphylococcus aureus is one of the main causes of community-acquired and hospital-acquired infections and is also one of the most common and deadly human pathogens (1). In addition to causing mild epidermal infections, it can also cause life-threatening systemic infections such as bacteremia, sepsis, toxic shock syndrome, meningoencephalitis, pneumonia, and endocarditis (2, 3). *S. aureus* possesses the ability to survive under harsh environmental conditions by producing a large number of extracellular virulence factors—such as hemolysin, coagulase, enterotoxin, protein A, and staphyloxanthin (STX)—and by forming biofilms (4).

In *S. aureus*, the predominant constituent of its total carotenoids is the C30 triterpenoid carotenoid α-D-glucopyranosyl-1-O-(4,4′-dimethoxy-sporene-4-carboxylate)-6-O-(12-methyltetradecanoate), designated as STX, which accumulates during the stationary phase (5, 6). As a critical virulence factor in over 90% of *S. aureus* strains, STX not only confers the characteristic golden pigmentation to the colonies but also, owing

Address correspondence to Fangyou Yu, wzjxyfy@163.com.

Jingyi Yu and Li Shen contributed equally to this article. The author order was determined based on their contribution to the article.

The authors declare no conflict of interest.

to its multiple conjugated double bonds, facilitates the detoxification of reactive oxygen species (ROS) (7) generated by the host immune system. Additionally, STX scavenges free radicals, singlet oxygen, hydrogen peroxide, and hypochlorous acid (8). The biosynthetic pathway of STX is initiated with the formation of dehydrosqualene from two molecules of farnesyl diphosphate, followed by a cascade of enzymatic reactions catalyzed by CrtM, CrtN, CrtP, CrtQ, and CrtO, culminating in the synthesis of STX (6). Coker et al. demonstrated that pigmented *S. aureus* strains exhibit enhanced resistance to neutrophil-mediated killing compared to non-pigmented strains in a murine model (9). Furthermore, STX contributes to the mechanical stabilization of the *S. aureus* membrane during infection and pathogenesis, presumably by modulating the biophysical properties of the membrane, thereby augmenting its rigidity (10, 11).

In the study conducted by Chen et al., methanol (MeOH) was employed for the extraction of pigment from *S. aureus*, and the quantification of pigment was determined by measuring the absorbance at 450 nm using a NanoDrop 2000c (Thermo Fisher Scientific) spectrophotometer (12). López et al. performed targeted metabolomics and biophysical analyses on both wild-type and knockout *S. aureus* strains to elucidate the biosynthetic pathways of STX and related carotenoids. In their methodology, methanol was utilized for repeated extraction of the pigment-containing supernatant, followed by a series of separation and purification steps. The extracted carotenoids were subsequently analyzed using high-performance liquid chromatography-diode array detection-atmospheric pressure chemical ionization tandem mass spectrometry (5). Barretto and Vootla isolated staphyloxanthin pigment from the resident microbiota of the midgut of healthy lepidopteran larva *Bombyx mori* and demonstrated that the staphyloxanthin pigment exhibits potent anticancer, antioxidant, antimicrobial, and DNA damage protection activity (13).

As a membrane-associated carotenoid, STX confers pathogenicity to the strain when intact within the cells; however, STX itself is a potent antioxidant. Therefore, this study focuses on evaluating the role of staphyloxanthin carotenoid pigments produced by bacteria in promoting biofilm formation and exerting antioxidant effects *in vitro*. We adopted the methodology described by Barretto and Vootla (13) and performed the initial extraction of STX using methanol. Notably, the methanol extract of *S. aureus* (MES) in this study contained additional components besides STX. However, we utilized methanol extracts from USA300 LAC *crtM* and USA300 LAC *crtN* mutants (staphyloxanthin-deficient *S. aureus* mutants) to demonstrate that STX within the MES was responsible for promoting biofilm formation. Our findings suggest a potential relationship between STX and biofilm formation. Since this study was conducted *in vitro*, further research is warranted to elucidate the roles of secondary metabolites extracted from microorganisms in mediating interactions among bacterial strains.

## RESULTS

### Effect of MES on the growth of *S. aureus*

Growth curves were plotted for clinical *S. aureus* isolates exposed to varying concentrations of the MES and methanol controls. As illustrated in Fig. 1, the 2.5% MES extract did not exhibit significant inhibitory effects on the bacterial proliferation of *S. aureus* clinical isolates, while 5% MES and methanol had an inhibitory effect on the growth of *S. aureus*. Tryptic soy broth (TSB) served as the negative control.

### Effect of MES on *S. aureus* biofilm development

Biofilm formation constitutes a critical virulence strategy in *S. aureus*, enabling immune evasion and antimicrobial resistance in host environments. To investigate MES-mediated modulation of biofilm formation, we employed a crystal violet-based semiquantitative assay coupled with confocal laser scanning microscopy (CLSM) analysis.

As depicted in Fig. 2A, MES treatment significantly enhanced biofilm biomass accumulation, evidenced by increased optical density at 600 nm ($OD_{600}$) values

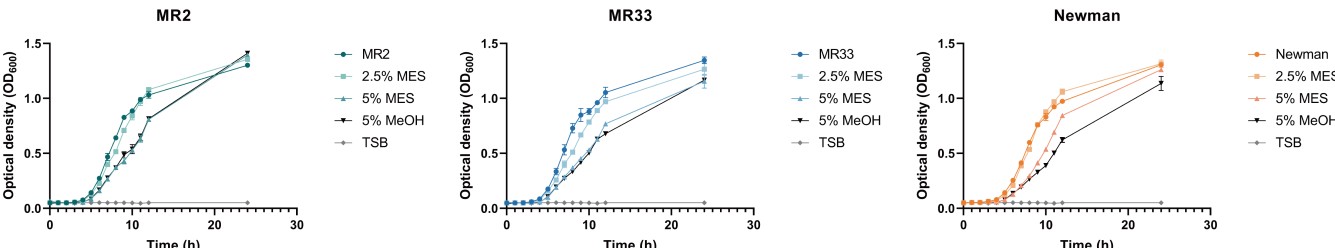

**FIG 1** Growth curves of *S. aureus* strains treated with MES. Strains were cultured in solutions containing 2.5%, 5.0%, 5.0% MeOH or no MES. Tryptic soy broth (TSB) served as a negative control. MeOH was also used to examine the effect of the solvent on bacterial growth. Each test was performed three times. Shown are representative results from a single experiment. Five percent MeOH represents sterile TSB containing a final volume of 5% methanol solvent, while 5.0% MES and 2.5% MES represent sterile TSB containing a final volume of 5.0% and 2.5% MES, respectively.

compared to untreated controls ($P < 0.05$). Solvent control experiments confirmed that MeOH exerted negligible effects on biofilm development (Fig. 2B). Notably, wild-type USA300 LAC strains exhibited superior ability to promote biofilm formation *in vitro* when treated with MES extracts compared to isogenic *crtM* and *crtN* mutants (Fig. 3), suggesting carotenoid biosynthesis pathway involvement in MES-mediated biofilm regulation. CLSM visualization revealed substantial biofilm enhancement in clinical methicillin-resistant *Staphylococcus aureus* (MRSA) isolates MR2, MR33, and Newman following 2.5% MES exposure (Fig. 4), and after 2.5% MeOH treatment, the biofilm was similar to the untreated group.

## Effects of MES on extracellular polymers

The extracellular polymeric substance (EPS) matrix is primarily composed of extracellular polysaccharides, proteins, and extracellular DNA (eDNA) (14). The enzymatic degradation of polysaccharides, DNA, and proteins was achieved using sodium periodate, DNase I, and protease K, respectively. As demonstrated in Fig. 5, treatment with 2.5% MES resulted in enhanced biofilm formation in both MR2 and Newman strains. However, post-treatment with DNase I and protease K attenuated the biofilm integrity of these two strains. Notably, MR33 exhibited differential susceptibility to these enzymatic interventions, where no significant reduction in biofilm density was observed following enzymatic digestion, though a marginal attenuation trend was evident (Fig. 5). The results demonstrated that the primary components of the biofilm formed by *S. aureus* following MES treatment were proteins and eDNA.

## MES increases the content of extracellular proteins in *S. aureus* biofilms

As shown in Fig. 5, MES increased the protein content in the biofilm of *S. aureus*. Therefore, investigating the effect of MES on extracellular protein secretion is crucial. As illustrated in Fig. 6, the extracellular protein content of MR2, MR33, and Newman following treatment with 25% MES was 1.72 times, 1.67 times, and 4.17 times that observed with 2.5% MeOH treatment, respectively, indicating that MES significantly promotes the synthesis and secretion of extracellular proteins.

## MES enhances the content of eDNA in the biofilm of *S. aureus*

eDNA is a critical nucleic acid component of biofilms and plays a pivotal role in the early stages of biofilm formation (15). The effect of MES on eDNA is illustrated in Fig. 7. Treatment with 2.5% MES resulted in an increased release of eDNA from *S. aureus*, indicating that MES significantly promotes eDNA release at this concentration. The extracellular DNA contents of MR2, MR33, and Newman after treatment with 2.5% MES were 1.31 times, 1.03 times, and 1.50 times that after treatment with 2.5% MeOH, respectively.

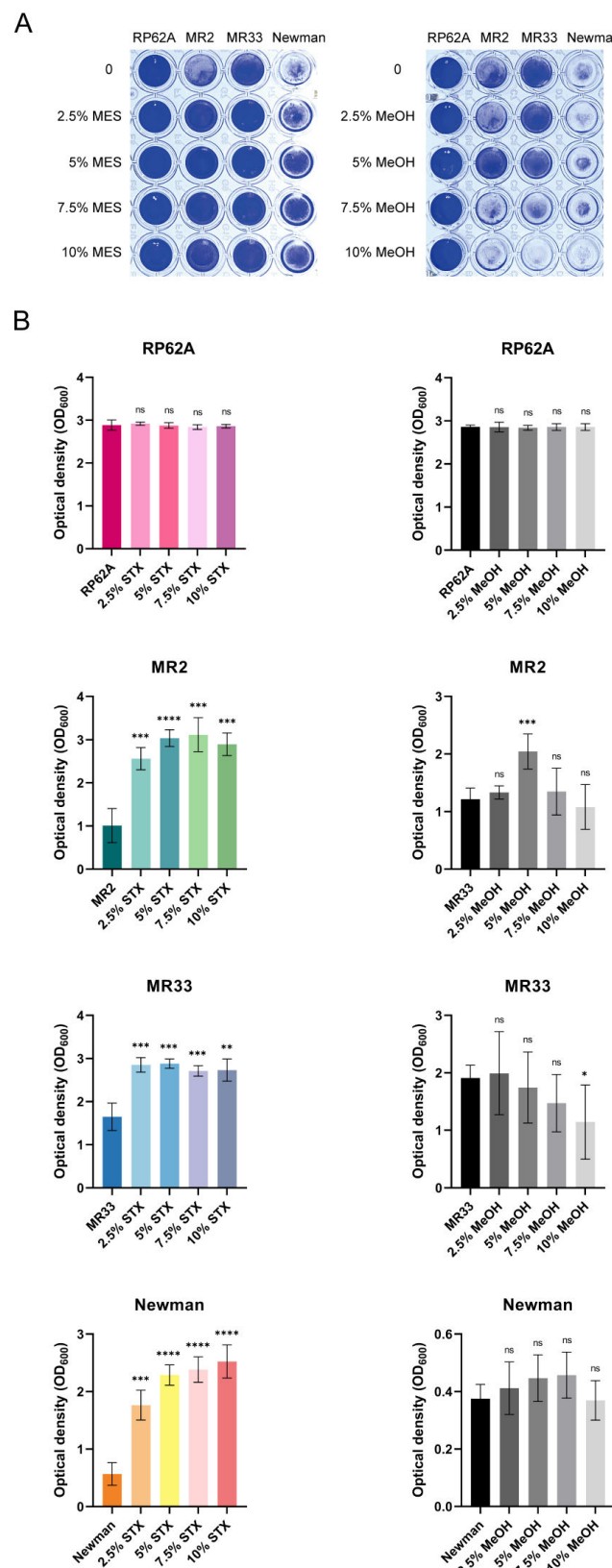

**FIG 2** Effect of MES on *in vitro* biofilm formation of *S. aureus*. (A) Biofilm growth of *S. aureus* in 96-well plates treated with MES and methanol at different concentrations. (B) The optical density (OD) value at 600 nm. RP62A is a standard strain of *Staphylococcus epidermidis* and serves as a positive biofilm control. *, $P < 0.05$; **, $P < 0.01$; ***, $P < 0.001$; ****, $P < 0.0001$. ns, no significance. Each test was performed independently in triplicate. The figure shows representative results of one experiment.

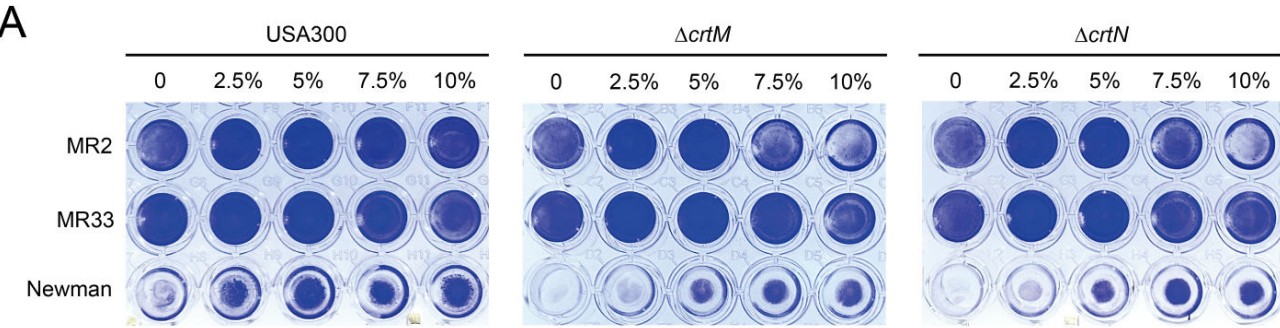

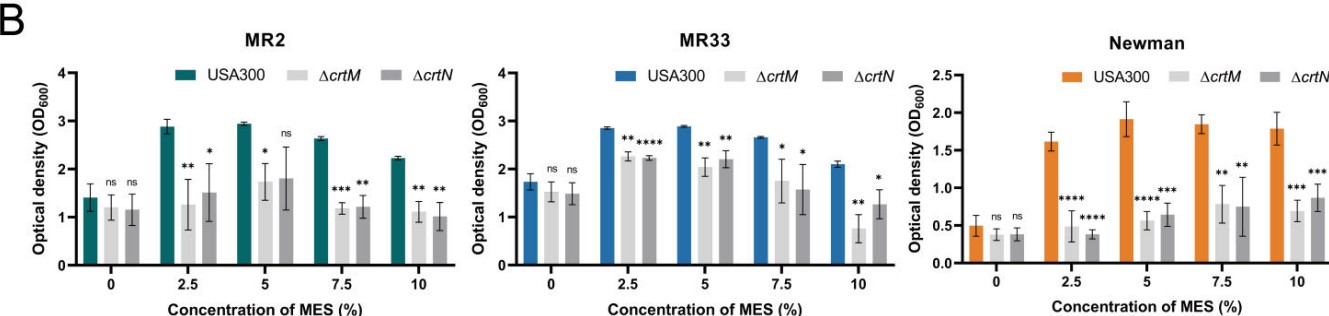

**FIG 3** *In vitro* assessment of biofilm formation in *S. aureus* following treatment with MES from USA300 LAC strain, *crtM* mutant, and *crtN* mutant of the USA300 LAC strain. (A) Representative images of biofilm development in 96-well microtiter plate assays. (B) Semiquantitative analysis of biofilm biomass using crystal violet staining. The absorbance at 600 nm ($OD_{600}$) was measured. *In vitro* evaluation of *S. aureus* biofilm formation following treatment with methanol extracts of the USA300 LAC strain, the *crtM* mutant, and the *crtN* mutant of the USA300 LAC strain. Δ*crtM* and Δ*crtN* represent methanol extracts of the *crtM* and *crtN* mutants of the USA300 LAC strain, respectively. *, $P < 0.05$; **, $P < 0.01$; ***, $P < 0.001$; ****, $P < 0.0001$. ns, no significance. Each test was performed independently in triplicate. The figure shows representative results of one experiment.

## MES enhances the antioxidant and whole blood killing abilities of *S. aureus*

Studies have shown that the main component of *S. aureus* STX is a C30 carotenoid membrane-bound pigment that can react with and inactivate ROS produced by macrophages and host neutrophils, thereby enhancing the resistance of *S. aureus* to immune clearance (16). We sought to determine whether the antioxidant capacity and resistance to whole blood killing of *S. aureus* changed with MES. Judging from the results in Fig. 8 and 9, after treatment with MES, the antioxidant capacity and antiwhole blood killing capacity of MR2 and Newman were significantly improved ($P < 0.01$), but the antioxidant capacity of MR33 did not change, and the antikilling ability of whole blood was enhanced ($P < 0.01$).

## MES induces oxidative stress in *S. aureus*

ROS, a major factor in antimicrobial activity, are generated when antimicrobial agents interact with bacterial solutions. In this study, ROS levels in bacteria were investigated. Increased fluorescence intensity reflects increased ROS levels. As shown in Fig. 10, after treatment with 2.5% MES, the fluorescence intensity of MR2 and MR33 increased ($P < 0.01$), indicating increased ROS levels. Therefore, it can be inferred that after treatment with 2.5% MES, MES may induce ROS levels, which then causes oxidative stress and triggers biofilm formation as a protective mechanism against the toxic effects of ROS. However, there was no significant change in the intracellular ROS level of Newman.

## Effects of MES on biofilm-related genes

To elucidate the molecular mechanism underlying MES-enhanced biofilm formation in *S. aureus*, we performed comparative transcriptional analysis of biofilm-associated genes

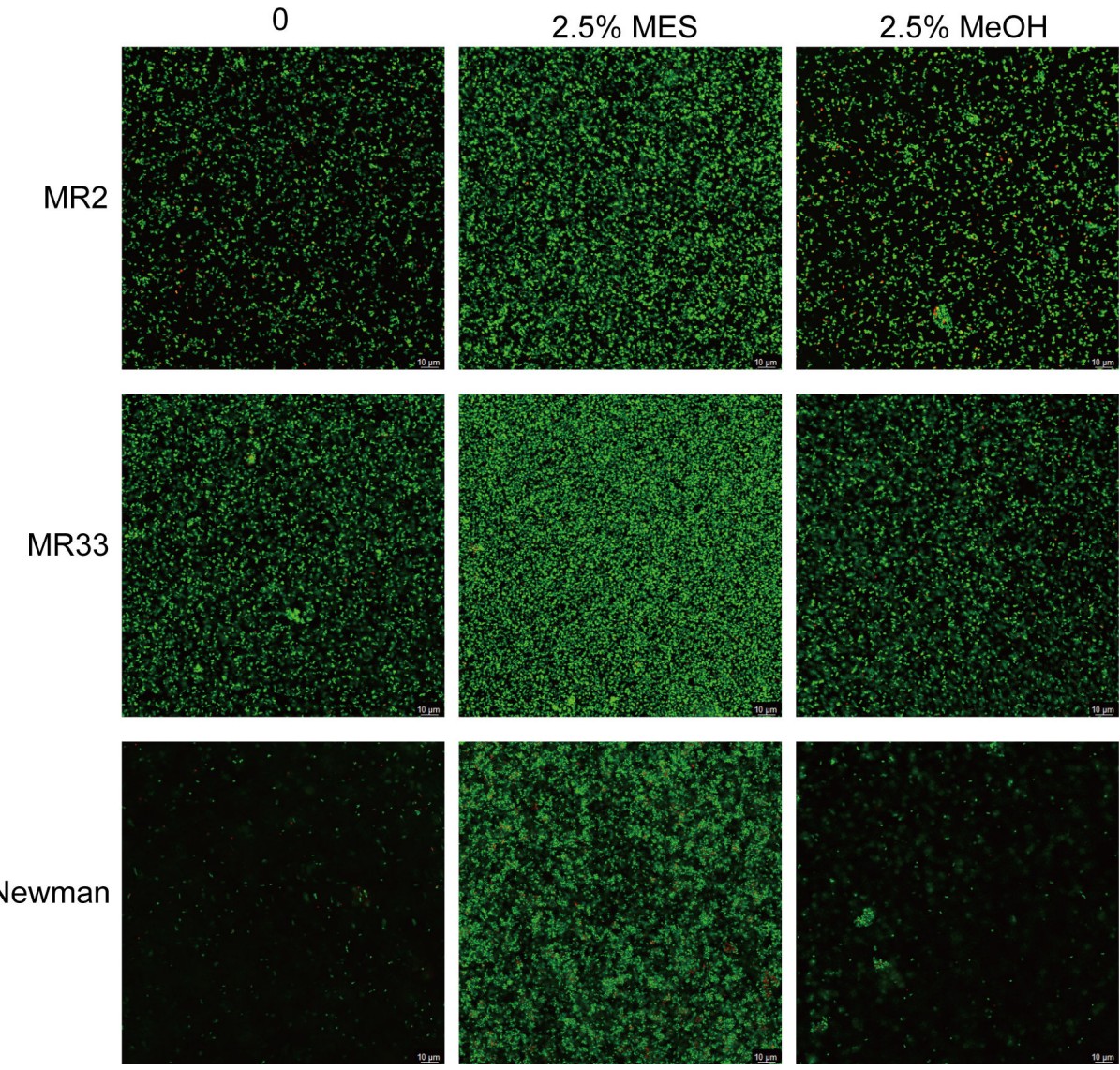

**FIG 4** CLSM observation of the effect of MES on *S. aureus* biofilm formation. Biofilms were stained with SYTO 9 and PI: SYTO 9 indicates all cells; PI indicates membrane-permeabilized cells. Biofilms not exposed to MES served as controls. A 2.5% MeOH group was used to eliminate the effects of the solvent on biofilms. Scale bar = 10 μm.

between MES-treated and untreated strains using quantitative reverse transcription PCR (RT-qPCR). Our findings revealed significant downregulation of key regulatory elements, including the two-component system gene *lytR* (which regulates bacterial autolytic activity) and its downstream effector gene *lrgA*. Conversely, genes encoding microbial surface components recognizing adhesive matrix molecules family adhesins demonstrated significant upregulation in MES-treated cells compared to untreated controls (Fig. 11). Specifically, we observed marked upregulation of fibronectin-binding protein A (*fnbA*)/*fnbB* (encoding fibronectin-binding proteins), serine-aspartate repeat-containing protein D (*sdrD*), clumping factors A/B (*clfA/B*), elastin-binding protein (*ebpS*), and fibrinogen-binding protein (*fib*).

## DISCUSSION

Marshall and Wilmoth (17) extracted the pigment from *S. aureus* S41 with methanol and identified the chemical structures of 17 intermediate compounds, all of which were

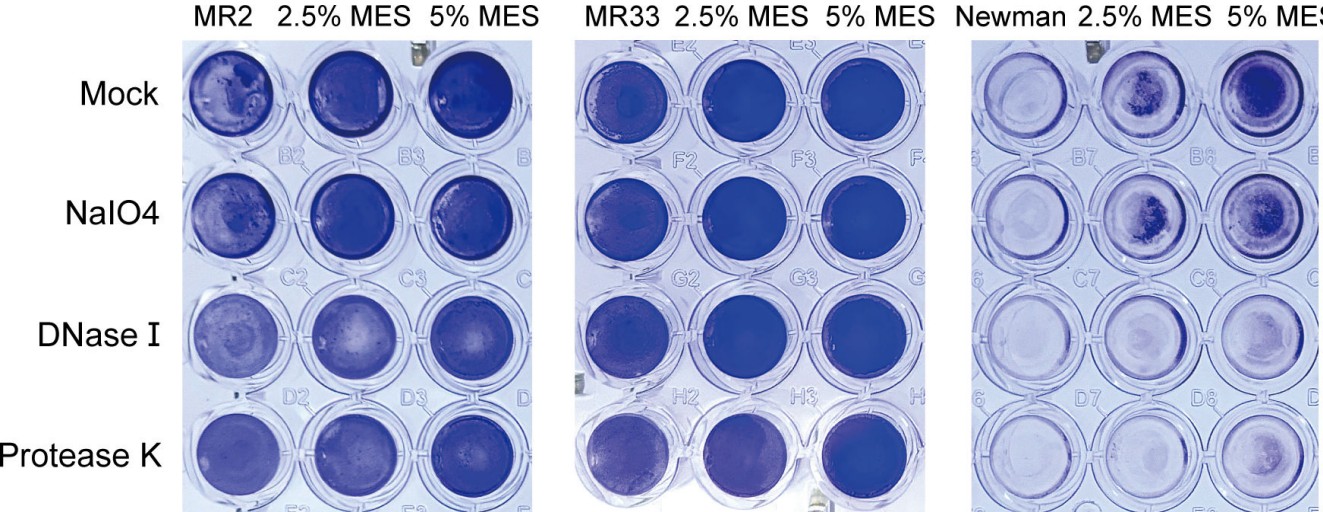

**FIG 5** Matrix degrading enzyme assay. The *S. aureus* biofilm treated with 2.5% MES and 5.0% MES was then treated with NaIO4, DNase I, and Protease K, respectively.

triterpenoid carotenoids with C30 structure, which was different from the common C40 carotenoid structure commonly found in most organisms. The main component of the pigment was staphyloxanthin, which was identified as α-D-glucopyranosyl-1-O-(4,4'-dia-poneurosporene-4-oate)6-O-(12-methyltetradecanoate). Pelz et al. overexpressed and purified staphyloxanthin in *Staphylococcus carnosus* (pTX-crt*OPQMN*) and analyzed it by high pressure liquid chromatography-mass spectrometry and nuclear magnetic resonance spectroscopy. Staphyloxanthin was identified as β-D-pyranosyl 1-O-(4,4'-dia-poneurosporene-4-oate)-6-O-(12-methyltetradecanoate) (6).

Studies have found that staphyloxanthin extracted from the silkworm intestinal microorganism *Staphylococcus gallinarum* KX912244 has potent anticancer, antioxidant, antibacterial, and DNA damage protection activities, which may be due to the potent antioxidant properties of the pigment (13). It also shows antibacterial activity against gram-positive bacteria *S. aureus*, gram-negative bacteria *Escherichia coli,* and *yeast Candida albicans* (13). The role of staphyloxanthin in oxidative defense is complex. ROS are a group of oxidatively active molecules generated during aerobic energy metabolism in cells. They play significant roles not only in the physiological processes of plants and animals but also in antibiotic bactericidal research and the development of bacterial resistance (18, 19). Depending on their sources and types, ROS exert different effects on bacteria. For instance, ROS produced by plants and animals in response to bacterial infections can exert bactericidal effects on pathogenic microorganisms, whereas ROS generated during bacterial growth and proliferation can ensure bacterial survival and promote biofilm formation (20).

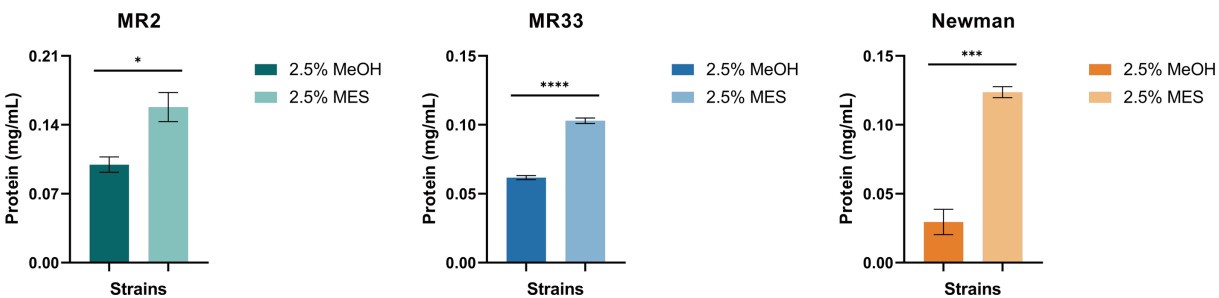

**FIG 6** Effect of MES on extracellular proteins in *S. aureus* biofilms. *, $P < 0.05$; ***, $P < 0.001$; ****, $P < 0.0001$. Each test was performed independently in triplicate. The figure shows representative results of one experiment.

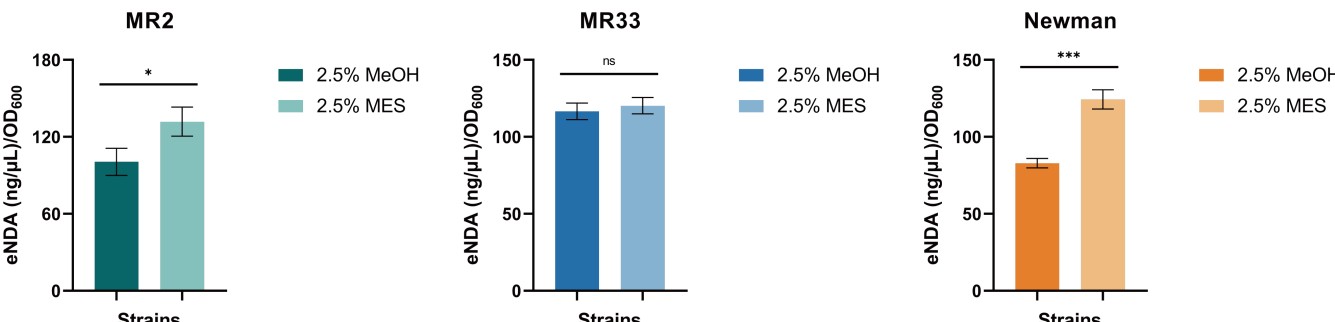

**FIG 7** The impact of MES on eDNA. *, $P < 0.05$; ***, $P < 0.001$; ns, no significance. Each test was performed independently in triplicate. The figure shows representative results of one experiment.

The results of some studies have shown that the LytSR two-component regulatory system plays an important role in the development of biofilms in *S. aureus*. LytSR positively regulates the transcription of *lrgAB* (21–23). The biofilms formed by the *lytS* mutant are more adhesive than those of the wild type and the complemented strain (21). In addition, the two-component system LytSR is related to the signal transduction of cell membrane potential perturbations and is involved in the adaptation of *S. aureus* to cationic antimicrobial peptides (24). Our study found that exposure to the MES significantly augmented the biofilm-forming capacity of *S. aureus*, coinciding with transcriptional repression of *lytR* and *lrgA*. Notably, RT-qPCR analysis demonstrated significant upregulation of key biofilm-associated genes, including *fnbA*, *sdrD*, clumping factors A/B (*clfA/B*), *ebpS*, and *fib*. Furthermore, enzymatic digestion assays demonstrated a substantial reduction in structural stability of MES-enhanced biofilms following treatment with DNase I and protease K, indicating a compositional shift in EPS dominated by enhanced DNA-protein matrix interactions. Studies have shown that DNase I can degrade extracellular DNA in the biofilm matrix and inhibit or disperse *Pseudomonas aeruginosa* and *S. aureus* biofilm (25, 26). Proteases can also degrade the extracellular matrix, leading to the release of planktonic cells and their components (27). Our study found that MES enhances biofilms by increasing the secretion of extracellular proteins of *S. aureus*. STX, produced by the *crt* operon of *S. aureus*, promotes resistance of *S. aureus* to oxidative stress and neutrophil-mediated killing (28).

After extracting the MES from *S. aureus*, we treated MR2, MR33, and Newman strains (which exhibit relatively weak STX production capacity) with these MES. Subsequent analysis revealed a substantial enhancement in both hydrogen peroxide resistance and whole blood survival capabilities among the aforementioned *S. aureus* strains. Liu et al. found that USA300 protected PAO1 from $H_2O_2$-mediated killing in co-culture, and this protection was dependent on STX (28). Since *S. aureus* and *P. aeruginosa* can colonize in the same niche *in vivo*, a few microns away from each other (29, 30), they hypothesized that this would form a low ROS groove around *S. aureus* cells, and this unique

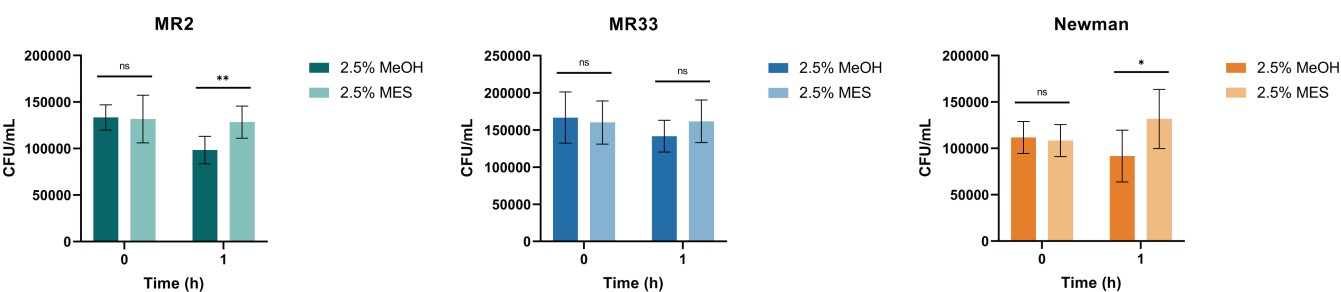

**FIG 8** MES enhances the ability of *S. aureus* to resist hydrogen peroxide *in vitro*. *, $P < 0.05$; **, $P < 0.01$; ns, no significance. Each test was performed independently in triplicate. The figure shows representative results of one experiment.

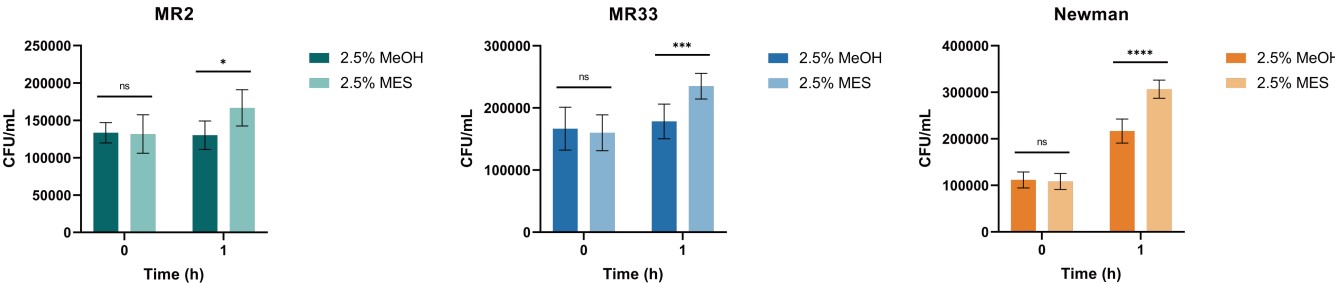

**FIG 9** MES enhances the ability of *S. aureus* to resist killing by whole blood *in vitro*. *, $P < 0.05$; ***, $P < 0.001$; ****, $P < 0.0001$. ns, no significance. Each test was performed independently in triplicate. The figure shows representative results of one experiment.

microenvironment might benefit neighboring *P. aeruginosa*. In summary, STX not only protects *S. aureus* but also protects *P. aeruginosa* from $H_2O_2$- and neutrophil-mediated killing (28). Studies have shown that ROS is an indispensable factor involved in the regulation of bacterial biofilm formation and virulence gene expression, and the ROS production of *S. aureus* is significantly increased during biofilm formation (31, 32). In this study, following the *in vitro* application of MES to *S. aureus* with weak biofilm-forming capabilities, an increase in intracellular ROS production was observed during the biofilm formation process. Consequently, we hypothesize that MES promotes biofilm formation in *S. aureus* by enhancing intracellular ROS generation, extracellular protein secretion, and eDNA release.

Our study still has some limitations. MES may contain a variety of metabolites.Using USA300 LAC *crtM* mutant and USA300 LAC *crtN* mutant can only provide indirect support for the role of STX, as it lacks purified compounds or chemical analysis (such as liquid chromatography–mass spectrometry). We will purify and analyze MES extracts in subsequent studies.

## Conclusions

In summary, our study revealed the role of the methanol extract of *Staphylococcus aureus* in the biofilm formation of *S. aureus*. MES can promote the formation of *S. aureus* biofilm and enhance the ability of *S. aureus* with weak staphyloxanthin production to resist whole blood killing and antioxidant capacity. We have only drawn preliminary conclusions, and further experiments are needed to confirm our conclusions. Our results may reveal the potential relationship between staphyloxanthin and biofilm, and further research is needed to study the biofilm formation ability of staphyloxanthin-producing and non-staphyloxanthin-producing *S. aureus* after co-culture.

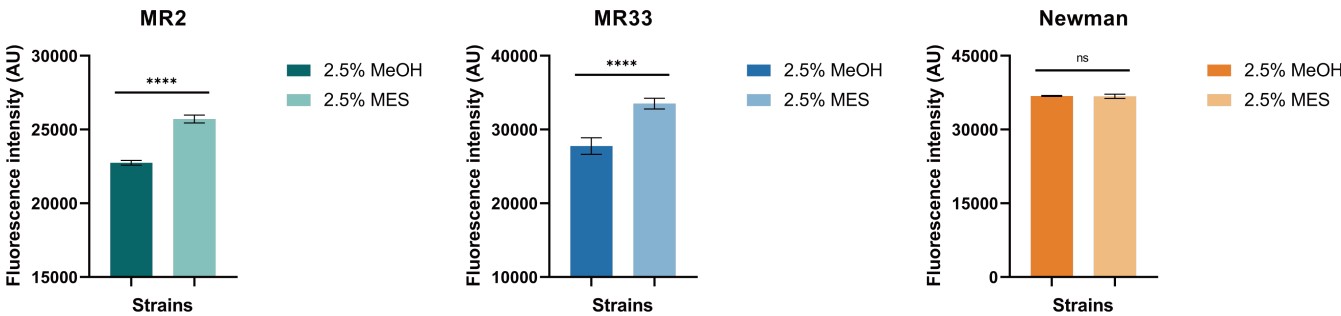

**FIG 10** Effects of MES on ROS levels in *S. aureus*. ****, $P < 0.0001$. ns, no significance. Each test was performed independently in triplicate. The figure shows representative results of one experiment.

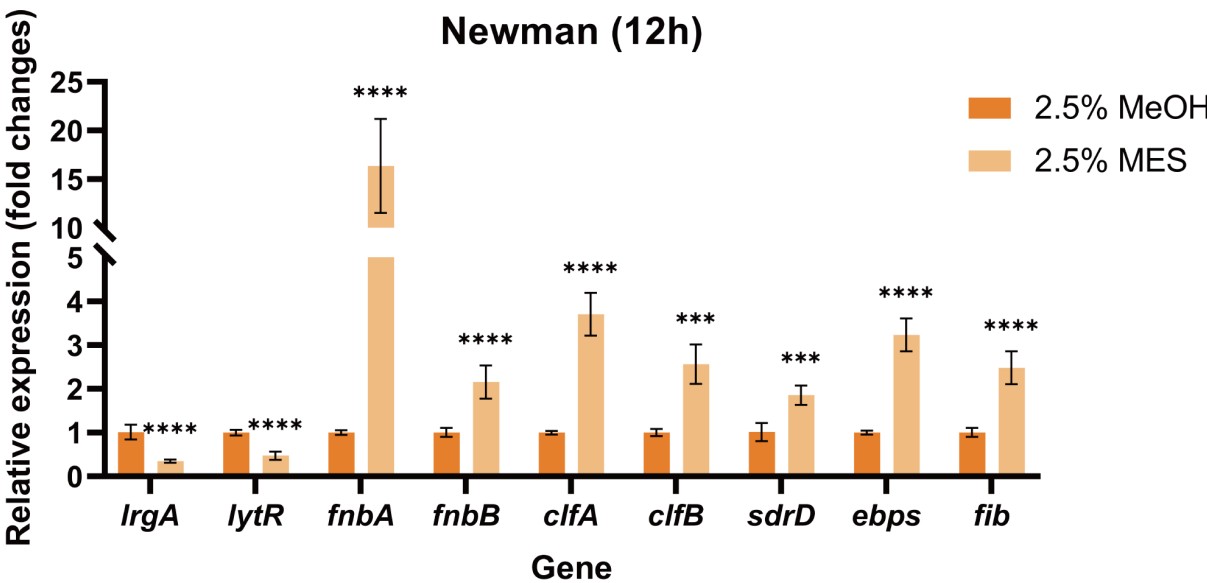

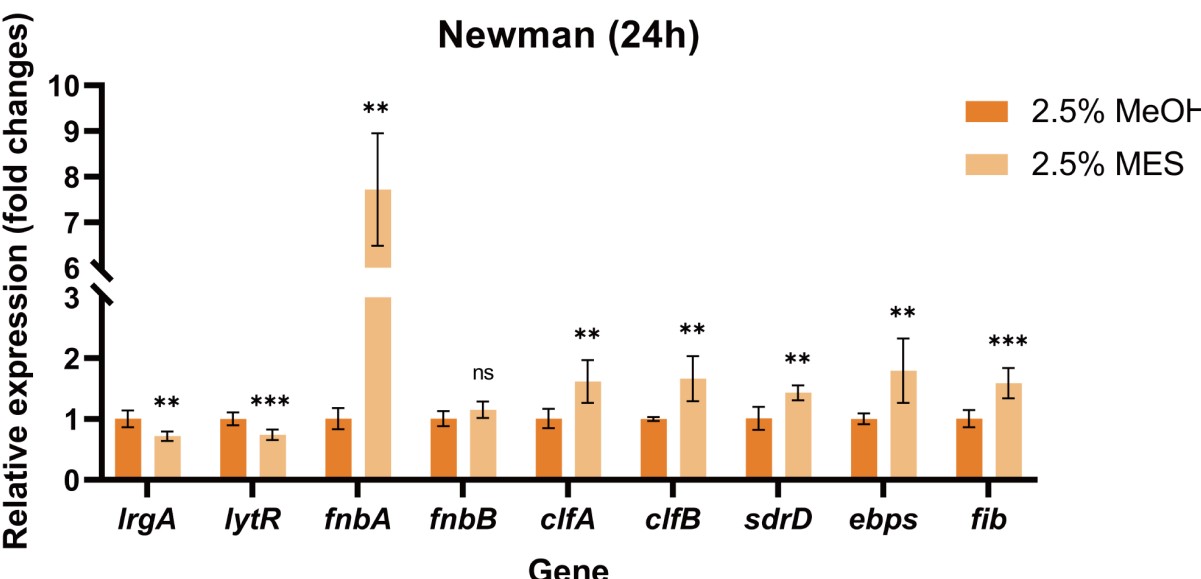

**FIG 11** The transcriptional expression levels of biofilm-associated genes in *S. aureus* following *in vitro* MES treatment were quantitatively assessed using reverse transcription-quantitative polymerase chain reaction at 12 and 24 hour time points. 2.5% MES and 2.5% MeOH represent Newman culture in sterile TSB containing 2.5% MES and 2.5% MeOH, respectively. **, $P < 0.01$; ***, $P < 0.001$; ****, $P < 0.0001$. ns, no significance. Each test was performed independently in triplicate. The figure shows representative results of one experiment.

## Limitations of the study

This study comprehensively investigated the potential of *Staphylococcus aureus* methanol extract to promote biofilm formation, as well as its antioxidant and antiwhole blood killing effects. However, several key limitations should be acknowledged. First, while we utilized crude methanol extracts of *S. aureus* membranes, including those from *crtM* and *crtN* knockout strains and the wild-type USA300 LAC, to demonstrate that STX is primarily responsible for enhancing biofilm formation, conclusive validation of these findings would require the use of purified STX. Second, all experiments in this study were conducted under *in vitro* conditions. Establishing a more complex experimental model

could provide deeper insights into the functions of secondary metabolites extracted from microbial strains and their interactions across different bacterial strains.

## MATERIALS AND METHODS

### Bacterial strains and growth conditions

The bacterial strains used in this study are listed in Table 1. MR33, MR2, Newman, and HG001 were used in this study. MR33 and Newman are methicillin-sensitive *Staphylococcus aureus* (MSSA); the standard laboratory strain HG001 is a derivative of NCTC8325 and also is an MSSA; and MR2 is an MRSA. MR2 and MR33 were gifts from the Laboratory Department of the Affiliated Hospital of Inner Mongolia Medical University. The standard strain of *Staphylococcus epidermidis* RP62A was donated by Professor Qu Di of Fudan University. Professor Lan Lefu and Professor Chen Feifei provided the USA300 LAC *crtM* mutant and USA300 LAC *crtN* mutant for our research. Unless otherwise stated, *S. aureus* strains were cultured in TSB (Oxoid) medium at 37°C with shaking at 220 rpm.

### Extraction of MES

Extraction of the MES was performed according to the previous method (35) with slight modifications. The bacteria were inoculated on Columbia blood agar plates and cultured overnight. All colonies on the plates were collected and washed twice with double-distilled water. After centrifugation, the pellet was suspended in 99% methanol and placed in the dark at 37°C with shaking at 220 rpm for 24 hours. The OD450 was measured after centrifugation at 8,000 rpm for 5 minutes.

### Growth assay

After overnight incubation of *S. aureus* at 37°C and 220 rpm, the suspension was adjusted to an optical density equivalent to the 0.5 McFarland standard. The bacterial suspension was then inoculated at a 1:100 ratio into 5 mL of TSB containing final concentrations of 2.5% MES, 5.0% MES, and 5.0% methanol, respectively. The cultures were incubated at 37°C with shaking at 220 rpm. The $OD_{600}$ was measured every hour for the first 12 hours and once at hours using a microplate reader (Wuxi Hiwell Diatek Instruments Co., Ltd.), and finally at the 24th hour. Each measurement was performed in triplicate.

### Effect on biofilm formation

Refer to the method of Peng et al. (36). *S. aureus* was inoculated into TSB and cultured overnight. The bacterial suspension was added at a 1:100 ratio to 200 µL of tryptic soy broth supplemented with 0.5% glucose (TSBG), and MES was incorporated to achieve final concentrations of 2.5%, 5.0%, 7.5%, and 10.0%, respectively. Correspondingly, controls containing equivalent concentrations of methanol (2.5%, 5.0%, 7.5%, and 10.0%) were prepared. The mixtures were incubated statically at 37°C for 24 hours. Then it was washed three times with 200 µL phosphate-buffered saline (PBS; Sangon Biotech [Shanghai] Co., Ltd.), fixed with 200 µL methanol for 15 minutes, stained with 50 µL 1% crystal violet for 8 minutes, rinsed with running water to remove the unstained crystal violet, dried, and dissolved with 200 µL of 30% glacial acetic acid, and the $OD_{600}$ was measured. The determination was repeated three times. We also extracted MES extract from the USA300 LAC *crtM* mutant and the USA300 LAC *crtN* mutant and repeated the above experiments.

### Enzyme hydrolysis experiment

The overnight culture of *S. aureus* was diluted 1:100 in TSBG, and MES was added to achieve final concentrations of 2.5% and 5.0% MES in TSBG, respectively. Subsequently, 200 µL of the mixture was transferred into each well of a 96-well polystyrene microtiter plate. After static incubation at 37°C for 24 hours, the supernatant was discarded. Then,

**TABLE 1** Strains used in this study

| Strain | Description | Reference or source |
|---|---|---|
| MR2 | Clinical strains | This study |
| MR33 | Clinical strains | This study |
| HG001 | Laboratory strain | (33) |
| Newman | Laboratory strain | (34) |

200 µL of PBS (as the control group), 40 nM sodium periodate, 1 µg/mL DNase, or 20 µg/mL protease K was added separately. After 2 hours, the supernatant was removed, and the wells were gently washed three times with 200 µL PBS (Sangon Biotech [Shanghai] Co., Ltd.) to eliminate unattached bacteria. Bacterial cells were fixed with 200 µL methanol for 15 minutes and 50 µL 1% crystal violet for 8 minutes. To remove planktonic cells, excess dye was rinsed with running water until the water became colorless. After drying, 30% acetic acid was added to dissolve the biofilm, and the $OD_{600}$ was measured using a microplate reader (Wuxi Hiwell-Detek Instruments Co., Ltd.). Each sample was tested in triplicate (37)

## Quantitative analysis of eDNA

eDNA isolation and quantification were performed as previously described (38, 39). *S. aureus* strains were cultured in six-well plates (Corning, New York, USA) as previously described and divided into groups containing 2.5% MES and 2.5% MeOH. Biofilms were resuspended in freshly prepared TES buffer (50 mM Tris-HCl, pH 8.0, 10 mM EDTA, 500 mM NaCl). Samples were centrifuged at 14,000 rpm for 5 minutes, and the supernatant was transferred to a new tube. An equal volume of phenol-chloroform-isoamyl alcohol (25:24:1) was added to the supernatant. The tubes were thoroughly mixed, and 3 vol of ice-cold anhydrous ethanol and 1/10 vol of glacial sodium acetate (3.0 M, pH 5.2) were added to the mixture. The mixture was then stored at −20°C overnight. eDNA was collected by centrifugation at 16,000 rpm for 20 minutes, washed with 75% ethanol, dried at room temperature, and finally dissolved in TE buffer. eDNA was quantified using a NanoDrop 2000 spectrophotometer (Thermo Fisher Scientific, USA). This experiment was performed in triplicate.

## Matrix protein quantification

According to the experimental protocol described by Stabile Gouveia et al. (40), the protein component of the biofilm matrix was estimated using Bradford reagent (B6916, Sigma-Aldrich). Overnight cultures of *S. aureus* were diluted 1:200 in TSBG (containing 2.5% MES and 2.5% MeOH, respectively) and incubated statically in six-well cell culture plates for 24 hours. The biofilm formed at the bottom of the wells was scraped and transferred to 1.5 mL Eppendorf tube. The biofilm was resuspended in washing buffer composed of 10 mM Tris-HCl (pH 8.0) and a protease inhibitor cocktail (Nacalai). The suspension was vortexed and centrifuged at 5,000 × $g$ for 10 minutes. The supernatant was transferred to a new tube. The pellet was dissolved in a matrix extraction buffer consisting of mM Tris-HCl (pH 8.0), 1 M NaCl, and a protease inhibitor cocktail, followed by incubation at 25°C for 30 minutes. After centrifugation at 5,000 × $g$ for 10 minutes, the supernatant was filtered through a 0.2 µm cellulose acetate filter (Sartorius Stim Biotech Pvt. Ltd., Göttingen, Germany). The filtered solution was used for protein quantification. The Bradford method was employed to quantify extracellular proteins present in the biofilm matrix: 100 µL of Bradford reagent was added to 400 µL of the filtered solution (as described above) and incubated at room temperature for 5 minutes. Absorbance was measured at 600 nm using a microplate reader. Each sample was analyzed in triplicate biological replicates.

## H$_2$O$_2$ killing assay

We followed the method of Chen et al. (12) with some modifications. The bacterial solution was added to 1 mL TSB (containing 2.5% methanol and 2.5% MES) in a 24-well plate at a ratio of 1:100. After 12 hours, the bacterial solution was adjusted to 0.5 McFarland and diluted 100 times. Five hundred microliters of the bacterial solution was mixed with 500 µL H$_2$O$_2$ (final concentration of 1 mM) and shaken at 250 rpm for 1 hour at 37°C. The mixture was diluted with PBS at 0 and 1 hour and spread on Columbia blood agar plates. The number of colonies (colony-forming unit [CFU]) was counted after overnight culture at 37°C.

### Intracellular detection of ROS

2',7'-Dichlorofluorescin diacetate (DCFH-DA) is an oxidation-sensitive fluorescent probe widely used for the detection of ROS. In the present study, the experimental procedure was performed as described in the protocol by Lin et al. (41). Briefly, *S. aureus* cultures, which had been pre-washed with sterile PBS, were mixed with DCFH-DA at a final concentration of 10 µmol/L at a volume ratio of 1,000:1. The mixture was then incubated at 37°C for 30 minutes.

Subsequently, extracellular DCFH was removed by centrifugation, followed by washing the bacterial pellets with sterile PBS. Afterward, the abovementioned bacteria were treated with extracted MES (2.5% by volume after addition) and the corresponding solvent MeOH. Following the treatment, the bacteria were resuspended in 1 mL of sterile PBS. For fluorescence intensity measurement, 200 µL of the resuspended sample was transferred into a 96-well microplate, and the change in fluorescence intensity of the bacterial suspension was detected using a fluorescence microplate reader. The detection parameters were set as follows: excitation wavelength at 485 nm and emission wavelength at 528 nm. Each experimental group was conducted in at least three independent replicates to ensure reproducibility of the results.

## Whole blood killing assay

The bacterial cells were subjected to overnight shaking, followed by inoculation of the bacterial suspension into 1 mL of TSB supplemented with 2.5% methanol and 2.5% MES at a 1:100 dilution ratio. The cell culture was incubated in a 24-well plate at 37°C for 12 hours, after which the bacterial suspension was collected and adjusted to 0.5 McFarland turbidity standard, followed by a 100-fold dilution. Subsequently, 500 µL of the bacterial suspension was mixed with an equal volume of human whole blood and incubated at 37°C with continuous shaking for 1 hour. The mixture was diluted with PBS at 0 and 1 hour time points and plated onto Columbia blood agar plates. Following overnight incubation at 37°C, CFUs were enumerated to quantify bacterial viability (12).

## qRT-PCR analysis

*S. aureus* strains were cultured in TSB containing 2.5% MES at 37°C, and bacterial cultures were harvested at 12 and 24 hours. A positive control consisted of a sterile tube inoculated with bacteria in 2.5% methanol. RNA was extracted according to the manufacturer's instructions (Spin Column Bacterial Total RNA Purification Kit; Sangon Biotech [Shanghai] Co., Ltd.). Next, cDNA was synthesized using the PrimeScript RT Kit with gDNA Eraser (TaKaRa, Tokyo, Japan) with 1 µg of extracted RNA per sample as a template. Quantitative real-time PCR was performed using TB Green Premix EX Taq (Tli RNaseH Plus) (TaKaRa) and a QuantStudio 5 Applied Biosystems (ABI) fluorescence quantitative PCR instrument (Thermo Fisher Scientific). Fold changes in gene expression were calculated using the comparative threshold cycle (Ct) method using the formula (2−ΔΔCt) to obtain RNA transcript levels of biofilm-related genes. Three biological replicates and three technical replicates were performed for each gene tested. The primers involved in qRT-PCR are listed in Table 2, with *gyrb* as the endogenous gene.

**TABLE 2** Primers used for real-time RT-PCR

| Primer | Sequence (5'–3') |
| --- | --- |
| *gyrB*-RT-F | ACATTACAGCAGCGTATTAG |
| *gyrB*-RT-R | CTCATAGTGATAGGAGTCTTCT |
| *lrgA*-RT-F | CTGGTGCTGTTAAGTTAG |
| *lrgA*-RT-R | GTATTGTTGAGACGATTATTAG |
| *lytR*-RT-F | AAGATGATAATAACGCAAGTG |
| *lytR*-RT-R | TAACGATTCAATGGCTCTG |
| *fnbA*-RT-F | TTCCTTAACTACCTCTTCT |
| *fnbA*-RT-R | CAATCATATAACGCAACAG |
| *fnbB*-RT-F | GCGAAGTTTCTACTTTTG |
| *fnbB*-RT-R | CAACCATCACAATCAACA |
| *clfA*-RT-F | CAGCGATTCAGAATCAGA |
| *clfA*-RT-R | GGCGGAACTACATTATTG |
| *clfB*-RT-F | CTGAGTCACTGTCTGAATC |
| *clfB*-RT-R | CTCAGACAGCGATTCAGA |
| *sdrD*-RT-F | CTATCTGAGTCTGAGTCT |
| *sdrD*-RT-R | TAATGGCTACTACGAAGA |
| *ebpS*-RT-F | GTGTGATGATTCGACTTG |
| *ebpS*-RT-R | CAGGATACAATAGAGAATACG |
| *fib*-RT-F | GTGCTTTACGGTGTGTTG |
| *fib*-RT-R | CTGCTATTAGTTTAACGGTATCAA |

## Statistical analysis

Statistical analysis was performed by Prism 9.0 software (GraphPad, San Diego, CA, USA). Data were analyzed with Student's *t*-test. *P* values of <0.05 were considered statistically significant (*$P$ < 0.05, **$P$ < 0.01, ***$P$ < 0.001, and ****$P$ < 0.0001).

## ACKNOWLEDGMENTS

The authors are grateful to Shanghai Pulmonary Hospital. The authors sincerely thank Professor Lan Lefu and Professor Chen Feifei for providing the USA300 LAC *crtM* mutant and USA300 LAC *crtN* mutant for our research.

This study was supported by grants from the Natural Science Fund of China (82202587) covering each section of this study.

Jingi Yu. and Y.S. designed and performed this work, and J.Y. drafted the work and revised it critically for important content. Jinjin Yang, Y.H., and L.S. participated in the experimental design and data analysis, J.S. and L.S. organized and produced Fig. 1 to 11 and Tables 1 and 2. F.Y. approved the content for publication. All authors read and approved the final manuscript.

## AUTHOR AFFILIATIONS

[1]Department of Clinical Laboratory, Shanghai Pulmonary Hospital, School of Medicine, Tongji University, Shanghai, China
[2]Department of Laboratory Medicine, The First Affiliated Hospital of Wenzhou Medical University, Wenzhou, China

## AUTHOR ORCIDs

Jingyi Yu http://orcid.org/0009-0007-6187-9580
Fangyou Yu http://orcid.org/0000-0003-4924-2484

## AUTHOR CONTRIBUTIONS

Jingyi Yu, Data curation, Writing – original draft | Li Shen, Investigation, Methodology, Project administration | Jinjin Yang, Methodology | Junhong Shi, Data curation, Writing

– original draft | Yu Huang, Methodology, Project administration, Software | Yongpeng Shang, Data curation, Methodology | Fangyou Yu, Project administration, Supervision

## DATA AVAILABILITY

The data sets generated during the current study are available from the corresponding author upon reasonable request. Most of the data are included in this published article.

## ETHICS APPROVAL

This study was approved by the Ethics Committee of Shanghai Pulmonary Hospital. All the experimental methods in this article were carried out in accordance with relevant guidelines and regulations.

## ADDITIONAL FILES

The following material is available online.

### Open Peer Review

**PEER REVIEW HISTORY (review-history.pdf).** An accounting of the reviewer comments and feedback.

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
