## [Reviewer comments · Microbiology Spectrum]

Microbiology Spectrum

Staphyloxanthin-Enriched Extracts Promote Biofilm Formation and Oxidative Stress Resistance in *Staphylococcus aureus*

Jingyi Yu, Li Shen, Jinjin Yang, Junhong Shi, Yu Huang, Yongpeng Shang, and Fangyou Yu

Corresponding Author(s): Fangyou Yu, Shanghai Pulmonary Hospital of Tongji University School of Medicine

Review Timeline:

Submission Date:	April 3, 2025
Editorial Decision:	July 24, 2025
Revision Received:	October 1, 2025
Accepted:	October 10, 2025

Editor: Rosemary She

Reviewer(s): The reviewers have opted to remain anonymous.

Transaction Report:

DOI: <https://doi.org/10.1128/spectrum.00996-25>

Re: Spectrum00996-25 (**Methanol extract of *Staphylococcus aureus* promotes biofilm formation and enhances antioxidant killing ability in vitro**)

Dear Prof. Fangyou Yu:

Thank you for the privilege of reviewing your work. After review of your work, the decision is Modifications. Due to difficulty finding available peer reviewers, I am providing Editor comments below to serve as one of the reviews. Below you will find my comments, instructions from the Spectrum editorial office, and the reviewer comments.

Revision Guidelines

Sincerely,
Rosemary She
Editor
Microbiology Spectrum

Editor comments:

Please define abbreviations at first use, for example ROS (line 101), MES (line 105), etc.

Introduction- please add more background on *S. aureus* methanol extract and the relationship of MES to staphyloxanthin and the other topics discussed in lines 73-98. As it stands, it is unclear why MES was chosen for study.

Results:

Lines 127-131, please clarify if the TSB line in Figure 1 contains any organism and label appropriately. It appears to be a background control, whereas the Methods describes inoculating bacteria to TSB with 0% MES. Please include the growth

control in the figure.

Figure 8, include significance of the asterisks displayed in the top and bottom charts.

Discussion -Lines 238-240, the relevance of pigment usage by humans to the present study is limited. Please reframe the discussion to focus on STX and its functionality in Staph.

Materials and Methods

-In general, the language throughout this section requires much improvement. This section should describe what was performed and not be phrased in form of instructions.

-Lines 319, 321, 322: instead of "belong to" suggest stating that a strain "is a" methicillin-resistant SA.

-Lines 350-351, please include origin of the strains and mutants used in the study.

-Line 361, please add reference number for Chen et al study.

-Whole blood killing assay, lines 368-376: this section is incorrect. It seems to describe hydrogen peroxide study. Make sure this properly describes a whole blood killing assay.

-qRT-PCR analysis, lines 378-383: Clarify cells (Staph aureus?) shaken overnight (temperature, settings). Please provide details on the qRT-PCR protocol/parameters.

Reviewer #1 (Comments for the Author):

This manuscript presents an intriguing study on the role of staphyloxanthin-rich methanol extracts (MES) from *S. aureus* in promoting biofilm formation and oxidative stress resistance in vitro. The authors utilize a combination of mutant analyses, gene expression profiling, microscopy, and functional assays to support their conclusion that MES enhances biofilm production and may act in trans to modulate phenotypes in *S. aureus* strains with poor intrinsic biofilm-forming capacity.

The work is timely and of potential interest to the microbial pathogenesis community. However, several aspects require clarification, refinement, and greater mechanistic insight before the manuscript can be considered for publication.

Title and Framing

The title, as it stands, is descriptive but overly broad and lacks a mechanistic focus. Given the centrality of staphyloxanthin (STX) in the findings, the title should be revised to reflect its mechanistic importance. For example:

"Staphyloxanthin-Enriched Extracts Promote Biofilm Formation and Oxidative Stress Resistance in *Staphylococcus aureus*"

Such a framing better captures the novelty of the findings and aligns with the narrative presented in the manuscript.

Introduction

The introduction is informative but somewhat repetitive and overly reliant on general statements. A few areas for improvement: Avoid redundancy regarding the antioxidant role of STX and its protective effect against neutrophils; the same concept appears multiple times across different paragraphs.

Most importantly, the rationale for using methanol extracts, particularly from crtM and crtN mutants, should be more clearly stated up front. Why methanol extraction? How does this approach allow one to dissect the functional contribution of STX?

Results

Line 127-131: To better quantify the inhibitory effect of the MES treatment, calculate the area under the growth curve (AUC) and provide appropriate statistical analyses. Even in the absence of statistically significant results, the inclusion of numerical data would enhance the interpretability and reproducibility of the findings.

It is unclear why only the Newman and MR2 strains were included in Figure 3, while MR33, despite showing a similar MES-induced increase in biofilm formation in Figure 2, is omitted. Please clarify the rationale behind the strain selection and whether MR33 was tested but yielded different results. This omission affects the consistency of the experimental design and interpretation.

The data shown in Figure 3 raises concerns. Specifically, both the MR2 and Newman strains display a statistically significant reduction in biofilm formation at the 0% treatment condition for the Δ crtM and Δ crtN mutants compared to wild-type USA300. Since these are untreated controls, this difference requires explanation.

Figure 5 presents results from enzymatic treatments of MES-induced biofilms but lacks any quantitative representation of the data (e.g., OD600 values, statistical analysis). Including numerical data would greatly enhance the clarity and robustness of the conclusions drawn, particularly regarding the relative contributions of proteins and extracellular DNA to biofilm structure.

The statistical analysis section is minimal and does not report the number of replicates used for each experiment, whether normality was tested, or whether corrections for multiple comparisons were applied. Clear information on experimental replication and statistical rigor should be added, especially given the reliance on p-values to support key claims.

The increase in hydrogen peroxide resistance after MES treatment is attributed to enhanced antioxidant capacity. However, no

direct measurement of intracellular ROS levels, catalase/peroxidase activity, or oxidative damage markers is provided. The conclusions would be stronger if supported by such mechanistic data.

The authors show that MES enhances biofilm biomass, particularly in strains with weak biofilm-forming capacity. However, some key controls are underdescribed. Specifically:
Was MES normalized for total pigment or protein content across strains? This would be important for ensuring comparability.
Did methanol alone affect growth or biofilm formation? This is addressed in part, but the control data should be clearly displayed alongside MES conditions in all figures.

The enzymatic degradation experiments are a useful touch but would benefit from clearer interpretation. For instance:
The matrix disruption assays suggest an increased role of eDNA and proteins in MES-enhanced biofilms. Could the authors measure eDNA concentration directly or confirm the presence of proteinaceous components?

The RT-qPCR data are compelling and provide a mechanistic explanation for MES-enhanced biofilm formation. Still, a few refinements would improve clarity:
Confirm whether the observed transcriptional shifts correlate with changes at the protein level (e.g., Western blot for FnbA/B or CifA/B if available).
Explain whether changes in *lytR/lrgA* are expected to increase autolysis, and how this would mechanistically promote biofilm biomass.

The enhancement in hydrogen peroxide resistance and whole blood survival is interesting and aligns with previous literature on STX. However, it's important to show actual CFU values (not just relative changes) to contextualize the biological relevance.

Discussion

The discussion successfully connects the data to broader concepts in microbial pathogenesis but can be improved in several areas:

Reduce speculative statements unless supported by direct evidence. For example, the hypothesis that STX "forms a low ROS groove" is intriguing but should be clearly framed as speculative unless experimental support is provided.

Throughout the manuscript, the MES is assumed to act primarily via STX. However, as the authors themselves acknowledge, the extract likely contains multiple metabolites. While the use of $\Delta crtM$ and $\Delta crtN$ mutants provides indirect support for the role of STX, the lack of a purified compound or chemical analysis (e.g., LC-MS) limits the strength of this conclusion. This limitation should be discussed more explicitly.

Clarify whether STX effects are direct (membrane integration, signaling) or indirect (global transcriptional shifts, quorum sensing).

The conclusions could be more concise and better structured to reflect the actual findings, while avoiding overstatements about clinical implications.

Figures and Data Presentation

The figure captions require significant improvement for clarity and completeness. For instance, the abbreviations TSB and 5% MeOH are used but never defined in any legend. In Figure 2, the X-axis does not represent "strains" but rather different treatment conditions-this should be clearly indicated. Additionally, the RP62A strain appears in the figure but is neither mentioned in the methods nor discussed in the main text. If this refers to *Staphylococcus epidermidis* RP62A, its inclusion and relevance should be justified and properly integrated into the narrative.

CLSM images should be supplemented with 3D reconstructions or z-stack summaries if available.
Include full statistical details in each figure (e.g., exact p-values, statistical test used).

Language and Style

Improve overall fluency and scientific tone.

1 This manuscript presents an intriguing study on the role of staphyloxanthin-rich methanol extracts
2 (MES) from *S. aureus* in promoting biofilm formation and oxidative stress resistance in vitro. The
3 authors utilize a combination of mutant analyses, gene expression profiling, microscopy, and
4 functional assays to support their conclusion that MES enhances biofilm production and may act in
5 trans to modulate phenotypes in *S. aureus* strains with poor intrinsic biofilm-forming capacity.
6 The work is timely and of potential interest to the microbial pathogenesis community. However,
7 several aspects require clarification, refinement, and greater mechanistic insight before the
8 manuscript can be considered for publication.

9 10 **Title and Framing**

11 The title, as it stands, is descriptive but overly broad and lacks a mechanistic focus. Given the
12 centrality of staphyloxanthin (STX) in the findings, the title should be revised to reflect its
13 mechanistic importance. For example:

14 *"Staphyloxanthin-Enriched Extracts Promote Biofilm Formation and Oxidative Stress Resistance in*
15 *Staphylococcus aureus"*

16 Such a framing better captures the novelty of the findings and aligns with the narrative presented
17 in the manuscript.

18 19 **Introduction**

20 The introduction is informative but somewhat repetitive and overly reliant on general statements.

21 A few areas for improvement:

22 Avoid redundancy regarding the antioxidant role of STX and its protective effect against
23 neutrophils; the same concept appears multiple times across different paragraphs.

24 Most importantly, the rationale for using methanol extracts, particularly from *crtM* and *crtN*
25 mutants, should be more clearly stated up front. Why methanol extraction? How does this
26 approach allow one to dissect the functional contribution of STX?

27 28 **Results**

29 Line 127-131: To better quantify the inhibitory effect of the MES treatment, calculate the area
30 under the growth curve (AUC) and provide appropriate statistical analyses. Even in the absence of
31 statistically significant results, the inclusion of numerical data would enhance the interpretability
32 and reproducibility of the findings.

33
34 It is unclear why only the Newman and MR2 strains were included in Figure 3, while MR33, despite
35 showing a similar MES-induced increase in biofilm formation in Figure 2, is omitted. Please clarify
36 the rationale behind the strain selection and whether MR33 was tested but yielded different
37 results. This omission affects the consistency of the experimental design and interpretation.

38
39 The data shown in Figure 3 raises concerns. Specifically, both the MR2 and Newman strains display
40 a statistically significant reduction in biofilm formation at the 0% treatment condition for the
41 $\Delta crtM$ and $\Delta crtN$ mutants compared to wild-type USA300. Since these are untreated controls, this
42 difference requires explanation.

43
44 Figure 5 presents results from enzymatic treatments of MES-induced biofilms but lacks any
45 quantitative representation of the data (e.g., OD600 values, statistical analysis). Including
46 numerical data would greatly enhance the clarity and robustness of the conclusions drawn,
47 particularly regarding the relative contributions of proteins and extracellular DNA to biofilm
48 structure.

49

50 The statistical analysis section is minimal and does not report the number of replicates used for
51 each experiment, whether normality was tested, or whether corrections for multiple comparisons
52 were applied. Clear information on experimental replication and statistical rigor should be added,
53 especially given the reliance on p-values to support key claims.

54

55 The increase in hydrogen peroxide resistance after MES treatment is attributed to enhanced
56 antioxidant capacity. However, no direct measurement of intracellular ROS levels,
57 catalase/peroxidase activity, or oxidative damage markers is provided. The conclusions would be
58 stronger if supported by such mechanistic data.

59

60 The authors show that MES enhances biofilm biomass, particularly in strains with weak biofilm-
61 forming capacity. However, some key controls are underdescribed. Specifically:

62 Was MES normalized for total pigment or protein content across strains? This would be important
63 for ensuring comparability.

64 Did methanol alone affect growth or biofilm formation? This is addressed in part, but the control
65 data should be clearly displayed alongside MES conditions in all figures.

66

67 The enzymatic degradation experiments are a useful touch but would benefit from clearer
68 interpretation. For instance:

69 The matrix disruption assays suggest an increased role of eDNA and proteins in MES-enhanced
70 biofilms. Could the authors measure eDNA concentration directly or confirm the presence of
71 proteinaceous components?

72

73 The RT-qPCR data are compelling and provide a mechanistic explanation for MES-enhanced
74 biofilm formation. Still, a few refinements would improve clarity:

75 Confirm whether the observed transcriptional shifts correlate with changes at the protein level
76 (e.g., Western blot for FnbA/B or ClfA/B if available).

77 Explain whether changes in *lytR/lrgA* are expected to increase autolysis, and how this would
78 mechanistically promote biofilm biomass.

79

80 The enhancement in hydrogen peroxide resistance and whole blood survival is interesting and
81 aligns with previous literature on STX. However, it's important to show actual CFU values (not just
82 relative changes) to contextualize the biological relevance.

83

84 **Discussion**

85 The discussion successfully connects the data to broader concepts in microbial pathogenesis but
86 can be improved in several areas:

87 Reduce speculative statements unless supported by direct evidence. For example, the hypothesis
88 that STX “forms a low ROS groove” is intriguing but should be clearly framed as speculative unless
89 experimental support is provided.

90 Throughout the manuscript, the MES is assumed to act primarily via STX. However, as the authors
91 themselves acknowledge, the extract likely contains multiple metabolites. While the use of $\Delta crtM$
92 and $\Delta crtN$ mutants provides indirect support for the role of STX, the lack of a purified compound
93 or chemical analysis (e.g., LC-MS) limits the strength of this conclusion. This limitation should be
94 discussed more explicitly.

95 Clarify whether STX effects are direct (membrane integration, signaling) or indirect (global
96 transcriptional shifts, quorum sensing).

97 The conclusions could be more concise and better structured to reflect the actual findings, while
98 avoiding overstatements about clinical implications.

99

100 **Figures and Data Presentation**

101 The figure captions require significant improvement for clarity and completeness. For instance, the
102 abbreviations TSB and 5% MeOH are used but never defined in any legend. In Figure 2, the X-axis
103 does not represent “strains” but rather different treatment conditions—this should be clearly
104 indicated. Additionally, the RP62A strain appears in the figure but is neither mentioned in the
105 methods nor discussed in the main text. If this refers to *Staphylococcus epidermidis* RP62A, its
106 inclusion and relevance should be justified and properly integrated into the narrative.

107

108 CLSM images should be supplemented with 3D reconstructions or z-stack summaries if available.
109 Include full statistical details in each figure (e.g., exact *p*-values, statistical test used).

110

111 **Language and Style**

112 Improve overall fluency and scientific tone.

We sincerely thank the editor and all reviewers for their valuable feedback that we have used to improve the quality of our manuscript. The reviewer comments are laid out below in italicized font and specific concerns have been numbered.

Response to Editor's comments:

1. Please define abbreviations at first use, for example ROS (line 101), MES (line 105), etc.

Respond: We sincerely thank you for your valuable suggestions for this study. This was an oversight on our part. We have followed your suggestion and have defined the abbreviations where they are first used, for example in the Abstract and Introduction. For example, line 40 and line 80.

2. Introduction- please add more background on *S. aureus* methanol extract and the relationship of MES to staphyloxanthin and the other topics discussed in lines 73-98. As it stands, it is unclear why MES was chosen for study.

Respond: According to your suggestion, we have added and explained in lines 90-118 of the manuscript. Recent studies have revealed interspecies synergistic interactions leading to co-infections, where exposure to *Pseudomonas aeruginosa* (*P. aeruginosa*) extracellular product 2-heptyl-4-hydroxyquinoline N-oxide (HQNO) STX production in *S. aureus*. *P. aeruginosa* HQNO mediates polymicrobial interactions with *S. aureus* by inducing STX production, thereby enhancing its resistance to innate immune effectors such as H₂O₂ and neutrophils (PMID:36711503). This implies that most bacteria exist within polymicrobial communities and, generating unique compounds (PMID:37078696) that influence individual and collective fitness and virulence (PMID:24825893). As a membrane-associated carotenoid, STX confers pathogenicity to the strain when intact within the cells; however, STX itself is a potent antioxidant. Therefore, this study focuses on evaluating the role of staphyloxanthin carotenoid pigments produced by bacteria in promoting biofilm formation and exerting antioxidant effects in vitro.

Coker MS et al. demonstrated that pigmented *S. aureus* strains exhibit greater resistance to neutrophil killing compared to non-pigmented strains in a murine model (PMID:29614256). Studies have reported that the strain is pathogenic STX is intact within the cells, although STX itself is a potent antioxidant. Delicia Avilla Barretto et al. (PMID:29651173) suggested that Staphyloxanthin pigment is a potential therapeutic agent, particularly due to its anticancer properties. Therefore, we adopted the methodology described by Delicia Avilla Barretto et al. and the initial extraction of STX using methanol. Notably, the methanolic extract of *S. aureus* (MES) contained additional components besides STX. Thus, we utilized methanolic extracts from USA300 LAC *crtM* and USA300 LAC *crtN* mutants (staphyloxanthin-deficient *S. aureus* mutants) to demonstrate that STX within the MES was responsible for promoting biofilm formation in vitro.

We have revised the introduction section of the manuscript.

3. Results:

3.1 Lines 127-131, please clarify if the TSB line in Figure 1 contains any organism and label appropriately. It appears to be a background control, whereas the Methods describes inoculating bacteria to TSB with 0% MES. Please include the growth control in the figure.

Respond: Based on your feedback, we have revised at line 125-126, where we specified the use of sterile TSB as the background control. We appreciate your suggestions, which have helped us address these overlooked details.

3.2 Figure 8, include significance of the asterisks displayed in the top and bottom charts.

Respond: Thank you for your reminder. Regarding the asterisks in Figure 8, we have provided

additional explanations in lines 280-281. We have also added annotations to other experimental result figures, including those in lines 236-239, 243-244, and 258-260. We appreciate your suggestions, which helped us address these overlooked details.

4. Discussion-Lines 238-240, *the relevance of pigment usage by humans to the present study is limited. Please reframe the discussion to focus on STX and its functionality in Staph.*

Respond: Based on your feedback, we have reconsidered and concluded that the sentence should indeed be removed to focus more precisely on STX and its role in *S. aureus*. Consequently, we have deleted the content originally found in lines 238-240. We have also revised and improved the discussion section (lines 295-309).

5. Materials and Methods

5.1 *In general, the language throughout this section requires much improvement. This section should describe what was performed and not be phrased in form of instructions.*

Respond: We sincerely appreciate your valuable feedback. Extensive revisions have been made to the Materials and Methods section. We have enhanced the description of the experimental procedures to provide greater specificity and clarity. Since the journal does not allow highlighting of revised parts, we can only tell you here that we have thoroughly revised the Materials and Methods sections.

5.2 *Lines 319, 321, 322: instead of "belong to" suggest stating that a strain "is a" methicillin-resistant SA.*

Respond: Based on your feedback, we have revised the text by replacing "belong to" with "is a" in lines 382-384.

5.3 *Lines 350-351, please include origin of the strains and mutants used in the study.*

Respond: We appreciate your reminder. Considering your valuable feedback, we have supplemented the description of the sources of the strains and mutants in lines 386-389.

5.4 *Line 361, please add reference number for Chen et al study.*

Respond: We appreciate your reminder and have added the reference number for the study by Chen et al. in line 474.

5.5 *Whole blood killing assay, lines 368-376: this section is incorrect. It seems to describe hydrogen peroxide study. Make sure this properly describes a whole blood killing assay.*

Respond: We appreciate your reminder and have revised the description of the whole blood killing assay in lines 500-510 to ensure accuracy.

5.6 *qRT-PCR analysis, lines 378-383: Clarify cells (Staph aureus?) shaken overnight (temperature, settings). Please provide details on the qRT-PCR protocol/parameters.*

Respond: We appreciate your reminder and have supplemented the experimental methodology for qRT-PCR analysis in lines 511-525.

We sincerely thank the editor and all reviewers for their valuable feedback that we have used to improve the quality of our manuscript. The reviewer comments are laid out below in italicized font and specific concerns have been numbered.

Reviewer #1 (Comments for the Author):

This manuscript presents an intriguing study on the role of staphyloxanthin-rich methanol extracts (MES) from *S. aureus* in promoting biofilm formation and oxidative stress resistance in vitro. The authors utilize a combination of mutant analyses, gene expression profiling, microscopy, and functional assays to support their conclusion that MES enhances biofilm production and may act in

trans to modulate phenotypes in *S. aureus* strains with poor intrinsic biofilm-forming capacity. The work is timely and of potential interest to the microbial pathogenesis community. However, several aspects require clarification, refinement, and greater mechanistic insight before the manuscript can be considered for publication.

Title and Framing

1. *The title, as it stands, is descriptive but overly broad and lacks a mechanistic focus. Given the centrality of staphyloxanthin (STX) in the findings, the title should be revised to reflect its mechanistic importance. For example: "Staphyloxanthin-Enriched Extracts Promote Biofilm Formation and Oxidative Stress Resistance in Staphylococcus aureus" Such a framing better captures the novelty of the findings and aligns with the narrative presented in the manuscript.*

Respond: Thank you for your valuable suggestions. After careful consideration, we agree with your suggestion and have changed the title of the manuscript to Staphyloxanthin-Enriched Extracts Promote Biofilm Formation and Oxidative Stress Resistance in *Staphylococcus aureus*.

2. Introduction

*The introduction is informative but somewhat repetitive and overly reliant on general statements. A few areas for improvement: Avoid redundancy regarding the antioxidant role of STX and its protective effect against neutrophils; the same concept appears multiple times across different paragraphs. Most importantly, the rationale for using methanol extracts, particularly from *crtM* and *crtN* mutants, should be more clearly stated up front. Why methanol extraction? How does this approach allow one to dissect the functional contribution of STX?*

Respond: We appreciate your reminder and have revised the Introduction section to avoid redundant elaboration on the antioxidant effects of STX and its protective role in neutrophils, as well as to prevent the repetition of the same concept across different paragraphs.

Regarding the rationale for using methanol extracts (particularly from *crtM* and *crtN* mutants), Coker et al. demonstrated that pigmented *S. aureus* strains exhibit greater resistance to neutrophil killing in murine models compared to non-pigmented strains (PMID:29614256). While STX is pathogenic when intact in cells, it is also a potent antioxidant. Delicia Avilla Barretto et al. (PMID:29651173) reported, "The harvested cell was re-suspended in distilled water. The pigment was then extracted with 100% methanol by repeated centrifugation until the cell pellet becomes colorless" and "The present study results suggest that Staphyloxanthin acts as a potential therapeutic agent especially due to its anticancer property." To explore whether STX possesses additional functions, we conducted this study.

Additionally we referenced the "Staphyloxanthin production assay" section in Chen et al. (PMID:26780405), which states, "Staphyloxanthin was extracted thrice with methanol and added to a total volume of 1 mL." Thus, we used methanol for the initial extraction of *S. aureus*, defined as MES in the manuscript. Since the methanol extract of *S. aureus* may contain metabolites other than STX, and *crtM* and *crtN* influence the biosynthesis of STX (PMID:26780405), we employed methanol extracts from USA300 LAC *crtM* and USA300 LAC *crtN* mutants (STX-deficient *S. aureus* mutants) to demonstrate that STX in MES promotes biofilm formation in vitro.

3. Results

3.1 *Line 127-131: To better quantify the inhibitory effect of the MES treatment, calculate the area under the growth curve (AUC) and provide appropriate statistical analyses. Even in the absence of statistically significant results, the inclusion of numerical data would enhance the interpretability and reproducibility of the findings.*

Respond: Thank you for your valuable suggestions. However, the following is my understanding, and I welcome any criticism or correction. In medicine, AUC primarily refers to the area under the plasma concentration-time curve, used to assess drug exposure in the body, while growth curves focus on how microorganisms change over time. AUC is a pharmacokinetic parameter, and the two belong to different research areas. Lines 120-134 of the original manuscript showed the growth of MR2, MR33, and Newman strains in the presence of 2.5%, 5% MES, and 5% MeOH. The purpose was to illustrate that although 5% MES and 5% MeOH had an effect on strain growth, both 2.5% MES and 5% MES could promote biofilm growth. We consulted an article published in Microbiology Spectrum (PMID: 37166296), which also conducted a growth curve study. The experimental results did not calculate the area under the growth curve (AUC). Therefore, we believe that it is not necessary to present the area under the growth curve (AUC).

3.2 *It is unclear why only the Newman and MR2 strains were included in Figure 3, while MR33, despite showing a similar MES-induced increase in biofilm formation in Figure 2, is omitted. Please clarify the rationale behind the strain selection and whether MR33 was tested but yielded different results. This omission affects the consistency of the experimental design and interpretation.*

Respond: Based on your comments, we have decided to include the corresponding results for the MR33 strain in the manuscript. The updated Figure 3 shows that strains MR2, 33, and Newman, treated with MES extracts from the USA300 *crtM* and *crtN* mutants, exhibited less biofilm formation than strains treated with MES extracts from the wild-type USA300 LAC strain (lines 159-170). This suggests that STX in MES promotes biofilm formation in vitro.

3.3 *The data shown in Figure 3 raises concerns. Specifically, both the MR2 and Newman strains display a statistically significant reduction in biofilm formation at the 0% treatment condition for the $\Delta crtM$ and $\Delta crtN$ mutants compared to wild-type USA300. Since these are untreated controls, this difference requires explanation.*

Respond: Based on your reminder, we re-examined the experimental data for Figure 3, removed some outliers, and re-plotted Figure 3, as described in lines 158-170 of the manuscript. Panel A of Figure 3 presents the results of a single experiment, where variations in the extent of crystal violet washing during semi-quantitative biofilm assay were observed. Figure 3B represents a comprehensive analysis of data from multiple experiments. We appreciate your reminder and will pay closer attention to experimental data processing in future studies.

3.4 *Figure 5 presents results from enzymatic treatments of MES-induced biofilms but lacks any quantitative representation of the data (e.g., OD600 values, statistical analysis). Including numerical data would greatly enhance the clarity and robustness of the conclusions drawn, particularly regarding the relative contributions of proteins and extracellular DNA to biofilm structure.*

Respond: We sincerely thank you for your valuable suggestions. In conjunction with your suggestions in 3.7, we addressed questions 3.4 and 3.7 simultaneously by measuring eDNA and protein concentrations. The results of these measurements, listed in lines 196-221, demonstrate that MES promotes biofilm formation by facilitating the release of extracellular proteins and eDNA from *S. aureus*.

3.5 *The statistical analysis section is minimal and does not report the number of replicates used for each experiment, whether normality was tested, or whether corrections for multiple comparisons were applied. Clear information on experimental replication and statistical rigor*

should be added, especially given the reliance on p-values to support key claims.

Respond: We sincerely appreciate your reminder. We will include explanatory notes at the end of each experimental procedure and provide detailed descriptions following each figure legend, as indicated in lines 279-280 and lines 258-260. Additionally, we acknowledge our oversight and have supplemented the data statistical analysis with additional explanations in lines 526-529.

3.6 *The increase in hydrogen peroxide resistance after MES treatment is attributed to enhanced antioxidant capacity. However, no direct measurement of intracellular ROS levels, catalase/peroxidase activity, or oxidative damage markers is provided. The conclusions would be stronger if supported by such mechanistic data.*

Respond: In accordance with your recommendations, we conducted experiments to detect intracellular reactive oxygen species (ROS) and supplemented the relevant content in the manuscript, starting from line 481 -499. The results are presented in lines 246-255. As illustrated in Figure 10, compared to 2.5% MeOH, 2.5% MES significantly elevated intracellular ROS levels in MR2, MR33 cells, with statistically significant differences observed.

The study "Resveratrol inhibits the formation of *Staphylococcus aureus* biofilms by reducing PIA, eDNA release, and ROS production" (PMID: 40370837) published by Jinfei He et al. mentioned that "Reactive oxygen species (ROS) are a class of highly reactive substances formed after the electrons of ground-state oxygen molecules are acquired. In bacterial biofilms, ROS induces genetic variation, promotes cell death in specific biofilm regions, and regulates biofilm development (PMID: 37143145). Studies have shown that ROS is an indispensable factor involved in the regulation of bacterial biofilm formation and virulence gene expression, and the ROS production of *Staphylococcus aureus* is significantly increased during biofilm formation (PMID: 35688112; PMID: 39458266). Nicotinamide adenine dinucleotide phosphate (NADPH) is involved in regulating these processes by catalyzing the production of ROS (PMID: 36426943). Therefore, reducing ROS production is an effective approach to explore the inhibition of *Staphylococcus aureus* biofilm formation." This research result is consistent with our experimental results.

3.7 *The authors show that MES enhances biofilm biomass, particularly in strains with weak biofilm-forming capacity. However, some key controls are underdescribed. Specifically: Was MES normalized for total pigment or protein content across strains? This would be important for ensuring comparability.*

Did methanol alone affect growth or biofilm formation? This is addressed in part, but the control data should be clearly displayed alongside MES conditions in all figures.

Respond: Thank you for your question. We describe the extraction of MES in lines 391-398: "Bacteria were inoculated onto Columbia blood agar plates and cultured overnight. All colonies on the plates were collected and washed twice with double-distilled water. After centrifugation, the precipitate was suspended in 99% methanol and incubated at 37°C in the dark with shaking at 220 rpm for 24 hours. After centrifugation at 8000 rpm for 5 minutes, the OD450 was measured." The MES used in our experiments was determined by measuring the OD450, ensuring that the OD450 of the MES used in different experiments was consistent. In addition, we present in Figure 2 the growth of biofilms after using different amounts of methanol alone.

3.8 *The enzymatic degradation experiments are a useful touch but would benefit from clearer interpretation. For instance: The matrix disruption assays suggest an increased role of eDNA and proteins in MES-enhanced biofilms. Could the authors measure eDNA concentration directly or confirm the presence of proteinaceous components?*

Respond: Based on your suggestion, we have added experiments measuring eDNA and protein content in biofilms. The corresponding experimental content has been added to lines 196-221, respectively. Our results show that 2.5% MES treatment increases the production of eDNA and protein in MR2, MR33, and Newman biofilms, with statistical significance ($p < 0.05$). In MR33, eDNA showed an increased expression trend after 2.5% MES treatment, but this trend was not significant, indicating that both eDNA and protein played an enhancing role in MES-enhanced biofilms.

3.9 *The RT-qPCR data are compelling and provide a mechanistic explanation for MES-enhanced biofilm formation. Still, a few refinements would improve clarity:*

Confirm whether the observed transcriptional shifts correlate with changes at the protein level (e.g., Western blot for FnbA/B or ClfA/B if available).

*Explain whether changes in *lytR/lrgA* are expected to increase autolysis, and how this would mechanistically promote biofilm biomass.*

Respond: Thank you for your suggestion. As you noted, the RT-qPCR data in our study are convincing and provide a mechanistic explanation for the enhanced biofilm formation achieved by MES. Previous studies have shown that the primary function of *lytSR* is to regulate programmed cell death in *S. aureus* (PMID: 22221897), which is associated with *lrgAB*. Furthermore, *lytSR* senses changes in cell membrane potential and participates in the response of *S. aureus* to cationic antimicrobial peptides (PMID: 23733465). Studies have also observed that *lytSR* regulates *S. aureus* biofilm formation by affecting *lrgAB* (PMID: 19502411). Our results (Figure 11) show that, under 2.5% MES, the two-component system gene *lytR* (regulating bacterial autolysis) and its downstream effector gene *lrgA* were downregulated. Genes encoding adhesins from the microbial surface components that recognize adhesion matrix molecules (MSCRAMMs) family, such as *fnbA/B* (encoding fibronectin-binding proteins), *sdrD* (serine-aspartate repeat protein D), *clfA/B* (agglutination factors), *ebpS* (elastin-binding protein), and *fib* (fibrinogen-binding protein), were significantly upregulated. I believe this demonstrates that "staphyloxanthin-enriched extracts promote biofilm formation." I agree with your suggestion, but due to laboratory conditions and other factors, we will further explore the relevant mechanisms in future research.

3.10 *The enhancement in hydrogen peroxide resistance and whole blood survival is interesting and aligns with previous literature on STX. However, it's important to show actual CFU values (not just relative changes) to contextualize the biological relevance.*

Respond: Thank you for your suggestion. We have replaced Figures 8 and 9 (lines 222-245) to show actual CFU values (rather than just relative changes).

4. Discussion

The discussion successfully connects the data to broader concepts in microbial pathogenesis but can be improved in several areas: Reduce speculative statements unless supported by direct evidence. For example, the hypothesis that STX "forms a low ROS groove" is intriguing but should be clearly framed as speculative unless experimental support is provided. Throughout the manuscript, the MES is assumed to act primarily via STX. However, as the authors themselves acknowledge, the extract likely contains multiple metabolites. While the use of $\Delta crtM$ and $\Delta crtN$ mutants provides indirect support for the role of STX, the lack of a purified compound or chemical analysis (e.g., LC-MS) limits the strength of this conclusion. This limitation should be discussed more explicitly. Clarify whether STX effects are direct (membrane integration, signaling) or indirect (global transcriptional shifts, quorum sensing). The conclusions could be

more concise and better structured to reflect the actual findings, while avoiding overstatements about clinical implications.

Respond: Thank you for your suggestion. We have removed the hypothesis that STX "forms a low-ROS groove" in the Discussion section, adding the relevant discussion of our research (lines 343-351). The supplement states a limitation of our study. While the use of $\Delta crtM$ and $\Delta crtN$ mutants provides indirect support for a role for STX, purified compounds or chemical analysis (e.g., LC-MS) are lacking (lines 352-355).

5. Figures and Data Presentation

The figure captions require significant improvement for clarity and completeness. For instance, the abbreviations TSB and 5% MeOH are used but never defined in any legend. In Figure 2, the X-axis does not represent "strains" but rather different treatment conditions-this should be clearly indicated. Additionally, the RP62A strain appears in the figure but is neither mentioned in the methods nor discussed in the main text. If this refers to Staphylococcus epidermidis RP62A, its inclusion and relevance should be justified and properly integrated into the narrative. CLSM images should be supplemented with 3D reconstructions or z-stack summaries if available. Include full statistical details in each figure (e.g., exact p-values, statistical test used).

Respond: Thank you for your suggestions. We have added and revised the figures in the manuscript. Information about the *S. epidermidis* standard strain RP62A has been added to lines 154–155 and 386–387. The abbreviations "TSB" and "5% MeOH" have been defined in detail in lines 132–134. Figure 2 has been updated between lines 150 and 151. We have described the statistical methods in lines 526–529.

6. Language and Style

Improve overall fluency and scientific tone.

Respond: Thank you for your suggestions. We have made extensive revisions to the manuscript.

Re: Spectrum00996-25R1 (**Staphyloxanthin-Enriched Extracts Promote Biofilm Formation and Oxidative Stress Resistance in *Staphylococcus aureus***)

Dear Prof. Fangyou Yu:

Your manuscript has been accepted, and I am forwarding it to the ASM production staff for publication. Your paper will first be checked to make sure all elements meet the technical requirements. ASM staff will contact you if anything needs to be revised before copyediting and production can begin. Otherwise, you will be notified when your proofs are ready to be viewed.

Sincerely,
Rosemary She
Editor
Microbiology Spectrum